# The Microbiome Structure of the Symbiosis between the Desert Truffle *Terfezia boudieri* and Its Host Plant *Helianthemum sessiliflorum*

**DOI:** 10.3390/jof8101062

**Published:** 2022-10-10

**Authors:** Lakkakula Satish, Hana Barak, Guy Keren, Galit Yehezkel, Ariel Kushmaro, Eitan Ben-Dov, Varda Kagan-Zur, Ze’ev Barak, Yaron Sitrit

**Affiliations:** 1The Jacob Blaustein Institute for Desert Research, Ben-Gurion University of the Negev, Beer-Sheva 84105, Israel; 2Avram and Stella Goldstein-Goren Department of Biotechnology Engineering and The Ilse Katz Center for Meso and Nanoscale Science and Technology, Ben-Gurion University of the Negev, Beer-Sheva 84105, Israel; 3Applied Phycology and Biotechnology Division, Marine Algal Research Station, Mandapam Camp, CSIR—Central Salt and Marine Chemicals Research Institute, Tamil Nadu 623519, India; 4Department of Environmental Engineering, Ben-Gurion University of the Negev, Beer-Sheva 841050, Israel; 5The Ilse Katz Center for Nanoscale Science and Technology, Ben-Gurion University of the Negev, Beer-Sheva 84105, Israel; 6School of Sustainability and Climate Change, Ben-Gurion University of the Negev, Beer-Sheva 84105, Israel; 7Department of Life Sciences, Achva Academic College, M.P. Shikmim, Arugot 7980400, Israel; 8Department of Life Sciences, Ben-Gurion University of the Negev, Beer-Sheva 841050, Israel

**Keywords:** bacterial diversity, desert truffles, *Helianthemum* *sessiliflorum*, mycorrhiza helper bacteria, *Terfezia boudieri*

## Abstract

The desert truffle *Terfezia boudieri* is an ascomycete fungus that forms ect-endomycorrhiza in the roots of plants belonging to Cistaceae. The fungus forms hypogeous edible fruit bodies, appreciated as gourmet food. Truffles and host plants are colonized by various microbes, which may contribute to their development. However, the diversity and composition of the bacterial community under field conditions in the Negev desert are still unknown. The overall goal of this research was to identify the rhizosphere microbial community supporting the establishment of a symbiotic association between *T. boudieri* and *Helianthemum sessiliflorum*. The bacterial community was characterized by fruiting bodies, mycorrhized roots, and rhizosphere soil. Based on next-generation sequencing meta-analyses of the 16S rRNA gene, we discovered diverse bacterial communities of fruit bodies that differed from those found in the roots and rhizosphere. Families of Proteobacteria, Planctomycetes, and Actinobacteria were present in all four samples. Alpha diversity analysis revealed that the rhizosphere and roots contain significantly higher bacterial species numbers compared to the fruit. Additionally, ANOSIM and PCoA provided a comparative analysis of the bacterial taxa associated with fruiting bodies, roots, and rhizosphere. The core microbiome described consists of groups whose biological role triggers important traits supporting plant growth and fruit body development.

## 1. Introduction

Desert truffles are fungi belonging to Ascomycetes that produce edible hypogeous fruit bodies. The genera of *Terfezia* and *Tirmania* establish mycorrhizal associations with their host plants, mainly *Helianthemum* species, which reside in arid and semi-arid ecosystems [1,2]. The geographical distribution of the species mentioned above is restricted to desert areas, around the Mediterranean basin, and up to the Persian Gulf. *Terfezia boudieri*, the most common desert truffle, is a socioeconomically important fungus traditionally taken as a valuable food and medicine, which is highly appreciated in modern cuisine [3,4,5,6]. Desert truffles are a rich source of minerals, amino acids, fatty acids, proteins, carbohydrates, fiber, vitamins, terpenoids, sterols, and flavor compounds. Moreover, they are rich in anti-bacterial, anti-cancer, and antioxidant compounds, all of which positively affect human health [3,5,7,8,9]. Cultivating desert truffles in dry environments is a beneficial way of utilizing croplands that have been considered unfertile [10]. For the cultivation of desert truffles, the rhizosphere’s physiology and biology, including microflora and mesofauna, are essential [11]. A rising focus on desert truffles is also being driven by a mandate to develop new crops adapted to arid zones [12]. However, minimal research has been conducted on desert truffle cultivation, biology, and genetics. Under natural conditions, the formation of mycorrhizal symbiosis is vital for the plant’s survival, wherever in boreal forests, tropical forests, savannas, or deserts [13]. The mycorrhizal association improves water uptake, phosphor ion supply, and other minerals by the host plant [10,13]. *Terfezia* species are proficient in forming various types of mycorrhizal associations, including endo, extend, and ectomycorrhiza, under specific conditions [3,14,15,16]. Mycorrhizal plants grown under desert conditions exhibit better physiological performance than non-mycorrhizal plants, i.e., better water use efficiency, photosynthesis, respiration, and plant development [3,17,18]. Therefore, the maintenance of mycorrhizal symbiosis, although energetically demanding, is vital for the survival of the host plant and fungi.

The rhizosphere is a confined area of soil subjected to the influence of a plant’s root [19]. Plant growth-promoting rhizobacteria (PGPR) include a versatile group of bacteria that span many phyla and help foster plant growth and development in different ways. In natural habitats, plant growth largely depends on bacteria, saprophytes, and mycorrhizal fungi, all of which facilitate the cycle and mobilization of nutrients. Various mechanisms are known in which microorganisms exert beneficial activities that support plant development. The synergistic applications of microbes to plants contribute to balancing hormonal levels, nutrient uptake, adaptation in osmotic tuning, antioxidant mechanisms, plant stomatal conductivity, the release of volatile organic compounds, exo-polysaccharide production, and help in sustaining the physiological fitness of plants for improved carboxylation linked with the increased growth rate in abiotic stress [20,21]. For example, phosphorus, an important macronutrient, is absorbed by plant roots primarily as inorganic orthophosphate, which is insoluble in most soils [22]. Various types of phosphatases secreted by rhizosphere bacteria can increase the hydrolysis rate of organic phosphate to inorganics, consequently making it accessible to plants [10,23].

The physiology and development of the host plant and its accompanying fungus strongly depend on the associated microbiome under both natural and nursery conditions [24,25]. Plants host a fascinating diversity of microbes inside and around their roots, aka the microbiome [26], which aid in absorbing nutrients and procuring the necessary elements or regulating hormonal imbalance [27]. Plants secrete a substantial portion of photoassimilates to the rhizosphere, which feeds soil bacteria. In exchange, plants receive the bulk of the NO_2_, NH_4_, P, and K minerals metabolized through diverse soil bacteria [28]. There is considerable attention paid to identifying and understanding the host-associated microbiomes of desert truffles with certain deliberate functions in their host plants. Still, it remains unclear whether the selection of a robust and stable microbiome is possible. To date, the microbiomes of some species, *Tuber borchii* [29,30], *Tuber magnatum* [31], *Tuber aestivum* [32,33,34], *Tuber melanosporum* [35,36,37], *Tuber indicum* [38,39], and *Tuber pseudobrumale* [40] truffles, have been identified. Despite this, little is known about the microbiome structure and composition of desert truffles under mycorrhiza with the host plant. The genome sequences of several desert truffles have been reported [41,42]. Thus, it is likely that the identification of the accompanying microbiome structures will support the status of desert truffles as a scientific model for studying complex symbiotic associations under desert conditions.

To address the microbiome structure of the symbiosis between the desert truffle *T. boudieri* and its host plant, *Helianthemum sessiliflorum,* we determined the structure of bacterial communities in the rhizosphere soil of the Negev Desert and in the symbionts. The results demonstrate differences at the phylum and class levels. Furthermore, the data uncovered distinct correlations and dynamic patterns between certain rhizosphere bacteria.

## 2. Materials and Methods

### 2.1. Subjects and Sampling

*T. boudieri* fruit bodies, mycelia-associated host plant *H. sessiliflorum* roots, and rhizosphere soil samples were collected from the experimental plots at Ramat Negev Desert Agriculture Center (30°51’59.99” N 34°46’59.99” E), Israel. The experiment was conducted for three years, and the biome results are from two successive years. Samples were taken during the winter–spring time during February–March. The roots and fruit bodies were gently washed with sterile water to remove the surface-bound soil particles. Rhizosphere soil was collected around the main root of the host and compared to soil collected from the edge of the experimental plot in soil without the hosts’ roots. The soil for control was collected 3 m away from the plot in the area between the plots. The plot was irrigated from October until May for two hours twice a week by 1.6 L/h drippers, and irrigation was stopped from June to September. Except for the weeding, no other works were carried out, and fertilization was avoided.

### 2.2. Total Genomic DNA Extraction

DNA was extracted from 100 mg samples of fruit bodies, rhizosphere soil, and roots using the QIAprep Spin Miniprep Kit (Qiagen, Hilden, Germany) in accordance with the manufacturer’s instructions. DNA yields from each sample were quantified using NanoDrop (Thermo Scientific NanoDrop 1000, San Diego, CA, USA) and agarose gel electrophoresis.

### 2.3. Next-Generation Amplicon Sequencing

Total genomic DNA extraction of all three samples was submitted to the DNA services (DNAS) facility at the Research Resources Center at the University of Illinois at Chicago (UIC) for bacterial small subunit (16S) ribosomal RNA (rRNA) gene sequencing using the Illumina MiSeq platform. Prior to sequencing, the library preparation protocol (Illumina, San Diego, CA, USA) was implemented. During the first PCR, fragments of the V3–V4 regions of the 16S rRNA gene fragments were amplified using 341F/805R universal primers attached with 5’ linker sequences CS1 and CS2 (known as common sequences 1 and 2).

### 2.4. Basic Processing of Amplicon Sequence Data

Forward and reverse reads were merged using PEAR [43]. Merged reads were trimmed to remove ambiguous nucleotides and primer sequences based on a quality threshold of *p* = 0.01. Reads that lacked either a primer sequence or sequences less than 300 bp were discarded. Chimeric sequences were identified and removed using the USEARCH algorithm compared to the Silva v132 reference sequence database [44,45]. Amplicon sequence variants (ASVs) were identified using DADA2 [46]. The representative sequences for each ASV were then annotated taxonomically using the Naïve Bayesian classifier included with DADA2 using the Silva v132 training set [44,46].

### 2.5. Alpha Diversity Analyses

Shannon indices were calculated using the vegan library with the default parameters in R [47]. Prior to the study, the data were rarefied to a depth of 15,000 counts per sample. The resulting Shannon indices were then modeled with the sample covariates using a generalized linear model (GLM), assuming a Gaussian distribution. The significance of the model (ANOVA) was tested using the F-test. Post-hoc pairwise tests were performed using the Mann-Whitney test. Plots were generated in R using the ggplot2 library [48].

### 2.6. Beta Diversity/Dissimilarity Analyses

Bray-Curtis indices were calculated with default parameters in R using the vegan library [47]. Prior to analysis, the normalized data were square root transformed. The resulting dissimilarity indices were modeled and tested for significance with the example covariates using the ADONIS test. Additional comparisons of the individual covariates were also performed using ANOSIM. Plots were generated in R using the ggplot2 library [48].

### 2.7. Differential Analysis of Microbial Taxa

Differential analyses of taxa as compared with experimental covariates (i.e., stress group, butyric acid levels) were performed using the software package edge Ron raw sequence counts [49]. Prior to analysis, the data were filtered to remove any sequences that were annotated as chloroplast or mitochondria in origin, as well as to remove taxa that accounted for less than 0.1% of the total sequence counts. Data were normalized as counts per million. Normalized data were then fit using a negative binomial generalized linear model with experimental covariates, and statistical tests were performed using a likelihood ratio test. Adjusted *p* values (*q* values) were calculated using the Benjamini–Hochberg false discovery rate (FDR) correction [50]. Significant taxa were determined based on an FDR threshold of 5% (0.05). The differential abundance of taxa was identified using the linear discriminant analysis (LDA) effect size (LEfSe) algorithm.

### 2.8. Determination of Soil N, P, and K Levels

Soil samples were collected in the years 2017, 2018, and 2019 from dune soils 3 m away from the edge of the experimental plot for control and from the experimental plot planted with *H. sessiliflorum* plants mycorrhized with *T. boudieri*. Soil sampling was conducted at the end of the wet season, at a depth of 0–0.15 m. Soil nitrogen (N), measured as extractable ammonium (NH_4_^+^) and extractable nitrate (NO_3_^−^) by potassium chloride extractions, extractable phosphorus (P), measured by the Olsen method [51], and extractable potassium (K+), measured by flame photometry, as described previously [52].

## 3. Results

### 3.1. Bacterial Diversity Varied among T. boudieri Fruit Bodies, Ectomycorrhizal Roots, and Rhizosphere Soil Samples

To compare the species diversity between the microbiomes of the host plant, the fungus, and the soil, a total of 1.8 million raw Illumina reads (an average of 200 K reads per sample) were obtained from the high-throughput sequencing of 16S V3–V4 hypervariable regions. Alpha diversity indices were compared across the investigated samples to estimate sample-specific variations in bacterial diversity. Figure 1 represents the sample-wise alpha diversity indices, such as OTUs, Shannon, and Chao1 and Simpsons, as box plots. Further comparisons among the values of Chao1 and other diversity indices, such as Shannon (which represents diversity) and Simpson (which represents evenness), showed a parallel pattern, exhibiting highly diverse and even bacterial diversity in samples of roots and rhizosphere soils when compared to fruit samples (Figure 1). This indicates that the roots of mycorrhized plants and fruit bodies are characterized by lower diversity than soils in the vicinity.

In addition, to determine the diversity variations in the alpha diversity indices, we also evaluated the inter-sample diversity variations (beta diversity) using a non-phylogenetic distance metric (Bray-Curtis) on the NMDS plot. The NMDS ordination showed a distinct microbial community associated with each of the analyzed samples (Figure 2). It is also important to note here that the bacterial communities of root and rhizosphere soil showed closer similarity compared to fruit samples. The peridium maintains some mechanisms that reduce diversity and allow the development of a specific bacterial community.

### 3.2. Bacterial Community Composition of T. boudieri Fruit Bodies, Ectomycorrhizal Roots, and Rhizosphere Soil Samples

To gain a better understanding of the bacterial taxonomic groups that inhabit the analyzed samples, Illumina-generated amplicon reads were further allocated at various taxonomic ranks (phylum, class, order, family, and genus levels). Across the analyzed samples, a considerable portion of the reads were assigned at the phylum level. However, a small portion (<1% of total reads) could not be allocated to any of the described phyla found in the database (termed unassigned). The relative proportion of dominant bacterial phyla is represented in Figure 3a, and each constituted the relative abundance of >1% of the entire community in a sample. As can be seen in the figure, the bacterial community composition of all the samples is occupied mainly by similar bacterial phyla (e.g., Proteobacteria, Bacteroidetes, Actinobacteria, Verrucomicrobia, Planctomycetes, Acidobacteria, Chloroflexi, and Firmicutes). However, a clear difference in their proportions can be observed among the fruit, root, and rhizosphere samples. Fruit-associated bacterial communities varied considerably from those associated with the roots and rhizosphere samples. A significant portion of the fruit-associated bacterial community was dominated by only two phyla, Proteobacteria (>53%) and Bacteroidetes (>42%), whereas root-associated and rhizosphere soil-associated bacterial communities were ruled by Proteobacteria (with a relative proportion of >59% in root and >31% in rhizosphere samples) and Actinobacteria (>23% in root and >24% in rhizosphere samples). The proportion of other bacterial phyla, such as Verrucomicrobia, Planctomycetes, Epsilonbacteraeota, Acidobacteria, Chloroflexi, Firmicutes, Gemmatimonadetes, Patescibacteria, Armatimonadetes, and Thaumarchaeota, was significantly lower among the fungal fruit body compared to root and rhizosphere soil samples.

Analysis at higher taxonomic ranks, wherein classes that represented >1% abundance of the whole community in any of the analyzed samples, also revealed similar results, indicating that the sample-specific taxonomic variations are higher among fruit bodies than root- and rhizosphere soil-associated microbiota (Figure 3b,c). The fruit-associated bacterial community was mainly characterized by high *Bacteroidetes*, *Gammaproteobacteria,* and *Alphaproteobacteria*. In contrast, bacterial classes, such as ABY1, NC10, and *Gracilibacteria,* represented the dominant classes in root and rhizosphere soil samples.

The hierarchical clustering pattern of the samples at the class level showed adjacent clustering of the root and rhizosphere soil samples (Figure 3c). For instance, replicates of root and rhizosphere soil samples were found to be clustered closely. In contrast, an entirely distinct clustering was observed for the fruit samples. The clustering pattern observed in the heat map analysis supported the results of the Bray-Curtis-based NMDS analysis. This confirmed that the soil- and root-associated bacterial communities closely resemble each other compared to those associated with the fruit bodies.

### 3.3. Bacterial Community Composition of T. boudieri Fruit Bodies, Ectomycorrhizal Roots, and Rhizosphere Soil Samples

Linear discriminant analysis (LDA) identified a total of 10, 12, and 5 bacterial groups that were exclusively enriched in the rhizosphere, plant roots, and fruit-associated bacterial consortia, respectively, with an LDA threshold of ≥3.5 (Figure 4).

Bacterial groups, such as *Pirellula*, Gemmataceae, Tepidisphaerales, *Bacillus*, *Chloroflexia*, Thermomicrobiales, *Solirubrobacter*, *Pseudarthrobacter*, Acidimicrobiia, and *Thermoanaerobaculales,* were significantly more abundant in the rhizosphere samples. *Steroidobacter*, *Pseudomonas*, *Sphingomonas*, *Bradyrhizobium*, *Rhizobium*, *Devosia*, *Phenylobacterium*, *Streptomyces*, Micromonosporaceae, *Actinoplanes*, Microbacteriaceae, and *Microbacterium* in root samples, and *Lysobacter*, *Variovorax*, *Chryseobacterium*, *Ohtaekwangia,* and *Chitinophaga* in fruit samples. The relative proportion of dominant biomarker bacterial groups that were exclusively associated with fruit samples is shown individually in Figure 5. These included Chitinophagales, Cytophagales, and Varivorax.

### 3.4. Soil N and K Level Changes after Mycorrhiza Colonization

Soil samples collected over three years from the non-fertilized plot of *T. boudieri* mycorrhized *H. sessiliflorum* plants revealed substantial variations in NO_2_, NH_4_, and K proportions when compared to a control of dune soil (Table 1). NO_2_ level was below the detection limit in control and mycorrhized soil samples in the first year after planting; however, some increase was determined in the following two years, 0.73 and 0.66 mg Kg^−1^ in dune soil, respectively, and similar levels were detected in the mycorrhized planted area. The NH_4_ ratio between 2017 and 2019 was significantly higher in the mycorrhized soil (increased up to 12.7 mg Kg^−^^1^) than in the control dune soil (4.2 mg Kg^−1^). Likewise, the level of K increased in the control soil between 2017 and 2019, from 0.58 to 8.3 mg/L, while in the mycorrhized plot, the level was 2.5-fold higher (20.8 mg/L) compared to the control non-mycorrhized soil (8.3 mg/L).

## 4. Discussion

There is an increasing consensus that microbial communities play an essential role in mediating ecosystem processes. Applying high-throughput sequencing to profile the host plant—*T. boudieri* microbiome allows the identification of previously unrecognized symbiotic associations between them. In this study, we investigated the diversity and compositional differences in the bacterial communities found in fungal symbionts, rhizospheres, and host plant root microbiomes toward understanding the tripartite mechanism and symbiosis. Our data clearly indicate that the composition of the bacterial community is significantly different between plant-free soil, roots, and truffle fruit bodies (Figure 1 and Figure 2). However, two communities, the rhizosphere soil, and roots, show much higher similarity in composition between them compared to fruit bodies of the desert truffle *T. boudieri* (Figure 2). The fruit body bacterial community is enriched by three bacterial orders: *Chitinophagales*, *Cytophagales*, and *Variovorex* (Figure 5). These three groups may represent the fungal recruitment of a specific bacterial community containing activities that include fungus protection from pathogens and providing nutritional support for mycelia growth. *Chitinophagaceae* can degrade complex organic matter, such as chitin and cellulose [53], and show β-glucosidase activity [54]. It is conceivable that *Chitinofagales* may support mycelia growth by supplying nutrients originating from organic matter degradation and, at the same time, protect the fruit body from insects and other chitin-containing pathogens. The second enriched bacterial member belongs to the *Cytophagales* order (Figure 5). These chemoorganotrophs are important for remineralizers of organic materials into micronutrients. They may support both mycelial growth and plant nutrition [55]. The third enriched group is bacteria that belong to *Variovorex,* a group that is known to have growth-promoting activity by reducing plant stress, increasing nutrient availability, and inhibiting the growth of plant pathogens; many of these mechanisms relate to the species’ catabolic capabilities [56]. Another example is the Chryseobacterium, which is also enriched in the fruit body. It is known that some plant-associated strains are able to inhibit plant pathogenic fungi [57].

The rhizosphere’s composition and proportion of the microbiome may benefit plant development and ascocarp growth. Analyses of α-diversity based on Shannon indexes show that the bacterial abundance is remarkably high across rhizosphere samples compared to plant roots and fungi samples, as expected (Figure 1). In contrast, roots-specific bacterial taxa were probably enriched by inhibiting some taxa while supporting others (Figure 1 and Figure 2). In the fruit bodies of truffles, the diversity is even lower. Conceivably, it is possible that the fungus enriches bacterial taxa that possess antifungal and anti-bacterial activities to protect and secure the development of spores. Indeed, many *Chetinophagales*, *Cytophagales,* and *Virovorax* possess such activities. In the host plant’s roots, groups of Gracili bacteria known as symbionts or endophytes were enriched [58]. Other enriched bacterial groups in roots involve nitrogen fixation, mineral acquisition, and metabolism. Among them are *Devosia* [59], *Nitrospira, Rhizobium,* and *Bradyrhizobium* (Figure 4). Potassium-solubilizing bacteria are represented by *Pseudomonas* spp. [60]. The main contribution of mycorrhizal fungi to plant development is the P ion supply [13]. Therefore, it is interesting to note that the host roots are enriched with *Microbacteria* and *Pseudomonas* spp., which are capable of solubilizing inorganic phosphorus from insoluble compounds (Figure 4). The latter may indicate tight tripartite relations between the partners that are central to plant nutrition and health.

Although the mycorrhization of *Terfezia* species with Cistaceae has been efficiently studied [61,62], more information about their microbiomes will contribute to defining how to intensify ectomycorrhizal colonization standards. We found that the bacterial communities in fruit bodies were diverse and distinct from those in the roots and rhizosphere (Figure 2). This is probably because *Terfezia* possesses effective antibacterial activity [63]. A comparative study of microbiomes for eight truffle species, including *Kalapuya brunnea*, *Leucangium carthusianum*, *Terfezia claveryi*, *Tuber gibbosum*, *Tuber indicum*, *Tuber lyonii*, *Tuber melanosporum*, and *Tuber oregonense,* has recently been reported [64]. Remarkably, the bacterial diversity within the truffles was minimal, ranging from 2–23 OTUs, with just a single *Bradyrhizobium* OTU predominant among Tuber fruit bodies, regardless of geographical provenance, and not found in the other truffle genera examined [64]. Clearly, we found a much more diverse bacterial community [30,34,39,65,66].

It is critical to understand which microbial species are abundant in the rhizosphere microbiome and their roles in improving desert truffle cultivation and biomass. Root-nodulating bacteria of the genera *Azotobacter*, *Bradyrhizobium*, *Klebsiella*, *Kosakonia*, *Rhizobium, Sinorhizobium,* and colonization by arbuscular mycorrhizal fungi can represent a strong tripartite symbiotic association with legumes [29,67]. These organisms’ consortia aid in the absorption of diffusion-restricted nutrients, such as NO_2_, NH_4_, P, and K. A recent study reported that the 12 genera of alpha- and beta-Proteobacteria were found to have possible symbiotic N-fixation capabilities, similar to our findings [68] (Table 1). Barbieri et al. [31] detected considerable levels of *Bradyrhizobium*, *Rhizobium*, and *Sinorhizobium* species in *T. magnatum* throughout truffle maturation. These microbes were discovered to be crucial in thiophene volatile organic compound production during the sexual development of white truffles [30]. This indicates that microbiome composition has many significant developmental effects beyond simple mineral nutrition for the fungus. The bacterial genera Pseudonocardineae are closely linked to the ectomycorrhization of *T. aestivum* [32]. These rhizosphere microorganisms are mycorrhization helper bacteria and can alter the abundance and character of root exudates, as well as the gene expression of ectomycorrhizal fungi [69,70,71]. Furthermore, although pathogenic microorganisms varied according to the truffle species investigated, a core microbiome consisting of alpha *Proteobacteria* from the family Bradyrhizobiaceae appears to be consistent with all varieties investigated thus far [29,30,35].

Mutualistic microbes often increase the development and viability of their host plants in situations of poor fertility, sandy surfaces, and inadequate plant carbon supplies. Bacterial species, including *Alcaligenes*, *Arthrobacter*, *Azospirillum*, *Azotobacter*, *Bacillus*, *Burkholderia*, *Enterobacter*, *Klebsiella*, *Pseudomonas,* and *Serratia,* are among the microorganisms that have been shown to help plants grow and develop effectively through producing or changing the endogenous plant growth regulator ratios [72] and solubilization of mineral phosphates and other nutrients [25,73]. The most potential and widely reported PGPR genera associated with desert truffles include *Bacillus*, *Phaeoacremonium*, *Pochonia*, *Pseudomonas,* and *Varivorax* [25,74,75]. *Varivorax* species have been identified as PGPRs that contain 1-aminocyclopropane-1-carboxylate deaminase [76]. *T. borchii* species contained *Actinomycetes* and *Pseudomonads*, which were phenotypically described for their probable involvement in symbiotic and spore formation [74]. The related taxa, i.e., Chitinophagales, Cytophagales, and Varivorax discovered in this study, might help researchers better understand the tripartite mechanism of *T. boudieri*, *H. sessiliflorum,* and their microbiome to improve in vivo plant production, mycorrhizal soil fertility, and sporocarp generation (Figure 3).

Farming of desert truffles in dune lands throughout the Middle East has been carried out, but with only moderate success, there are several other undiscovered factors to consider in the better management of orchards prior to truffle production, including fertilizer usage. Here, we systematically evaluated the difference in NO_2_, NH_4_, P, and K levels before and after *T. boudieri* establishment, where K seems to have an essential role in plant-mycorrhiza development in these fields (Table 1). It is interesting to note that the levels of phosphor, which are considered the basis for the establishment of plant–mycorrhiza associations, did not change over the years (data not shown). Identifying suitable fertilizers to apply in productive truffle cultivation plots might be helpful to increase ascocarp production [77]. According to Sourzat [78], using fertilizers in the proper proportions boosted *T. melanosporum* productivity. However, similar studies also found that continual NO_2_, NH_4_ and P treatment in the farms reduced mycorrhizal colonization. This was most likely owing to a drop in host involvement in the fungal symbiont to scavenge soil nutrients and essential minerals [79,80]. Morte et al. [81] stated that desert truffle *T. claveryi* mycorrhizas increases *Helianthemum almeriense* sustainability over drought times by modifying functional and nutritional characteristics. The presence of *T. claveryi* in the fields boosted the absorption of NO_2_, NH_4_, P and K by *H. almeriense* plants [18]. It has been shown that ectomycorrhizal formation alters the microbiome architecture in the rhizosphere and on the roots of host plants [75]. Diazotrophs (nitrogen-fixing microorganisms) *Azotobacter* was identified as a significant nitrogen-fixing free-living microbe and as a possible bacterial biofertilizer with shown effectiveness for soil nutrients and organic soil fertility [82]. It is possible to replace or supplement mineral chemical fertilizer with biological nitrogen fixation, a bacteria-mediated process that converts atmospheric nitrogen into ammonia by susceptible bacterial enzymes [82,83]. Selosse et al. [84] stated that as part of the usual lifecycle from haploid spore germination to haploid free-living mycelium development in soil, the formation of a mutually supportive symbiotic relationship with the roots of host plants, as well as the formation and maturation of the fruiting body, is included [85].

The symbiotic relationship of *T. boudieri* with its host plant, *H. sessiliflorum*, results in significant growth of the host, probably affecting fruit body size and biomass. Enhanced plant growth supports increased ectomycorrhizal growth because the host’s physiological performance and that of the two symbiotic partners depend on the bacterial supply of nitrogen and potassium. Similarly, the host plant’s associated microbiome composition has also been shown to influence truffle growth and biomass increases [86]. However, how plants and fungi organize bacterial communities is not yet fully understood, and several mechanisms have been suggested, including chemotropism, which is still unknown for truffles. It is possible that desert truffles can manipulate the bacterial community by secreting molecules that possess antimicrobial activity [63]. The results of this and further studies will aid in understanding soil fertilization. Of particular importance is the promotion of systematic molecular comparisons and functional investigations.

## 5. Conclusions

In conclusion, obtaining more profound knowledge about the direct relations and contributions between the tripartite partners in this symbiosis will help us improve the synthetic symbiotic conditions necessary for establishing truffles as a new crop for arid zones. Inoculation of host plants in these nurseries should include not only mycorrhiza but also beneficial bacteria that will support the development of the host and the desert truffles.

## Figures and Tables

**Figure 1 jof-08-01062-f001:**
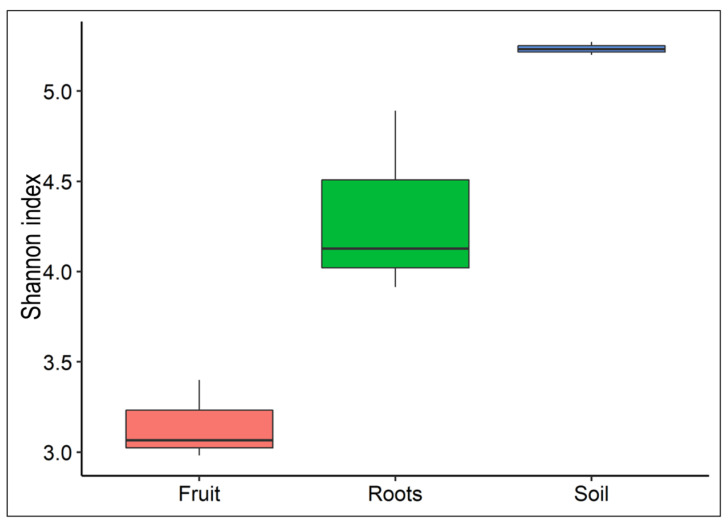
Differences in microbiome diversity of fruit bodies, mycorrhized roots, and rhizosphere. Alpha diversity was based on the Shannon indexes. (Mean values: Fruit: 3.14, Roots: 4.31, and Soil: 5.23).

**Figure 2 jof-08-01062-f002:**
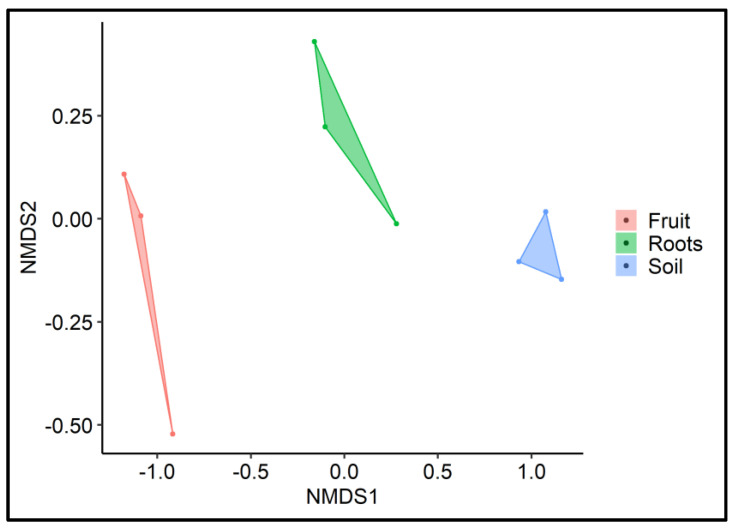
Inter-sample diversity variations of fruit bodies, mycorrhized roots, and rhizosphere. Non-metric multidimensional scaling (NMDS), based on Bray-Curtis, with convex hulls (Stress level: 0.02), was used.

**Figure 3 jof-08-01062-f003:**
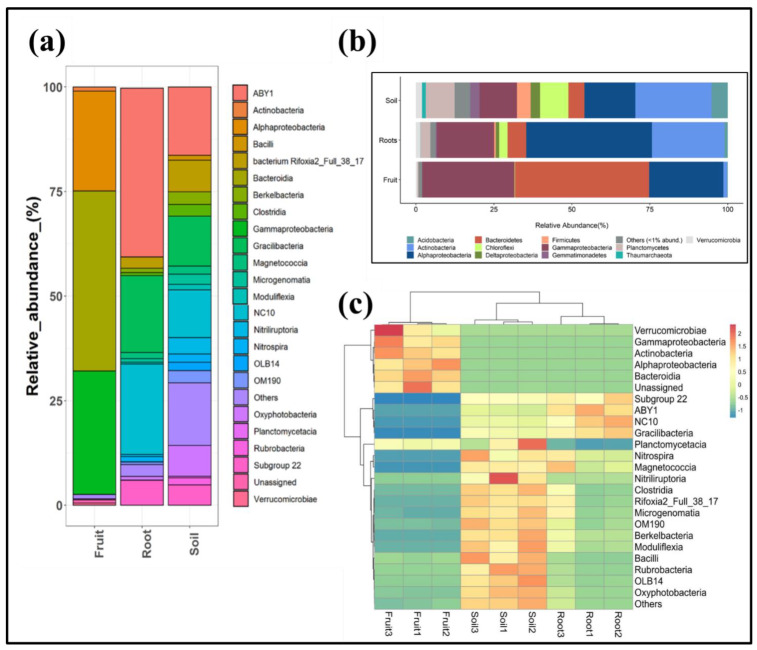
Relative abundance of dominant bacterial groups at the phylum level (**a**); Relative abundance of dominant bacterial groups at class level (**b**); Clustering pattern of the bacterial community (at class level) associated with fruit, root, and rhizosphere samples (**c**).

**Figure 4 jof-08-01062-f004:**
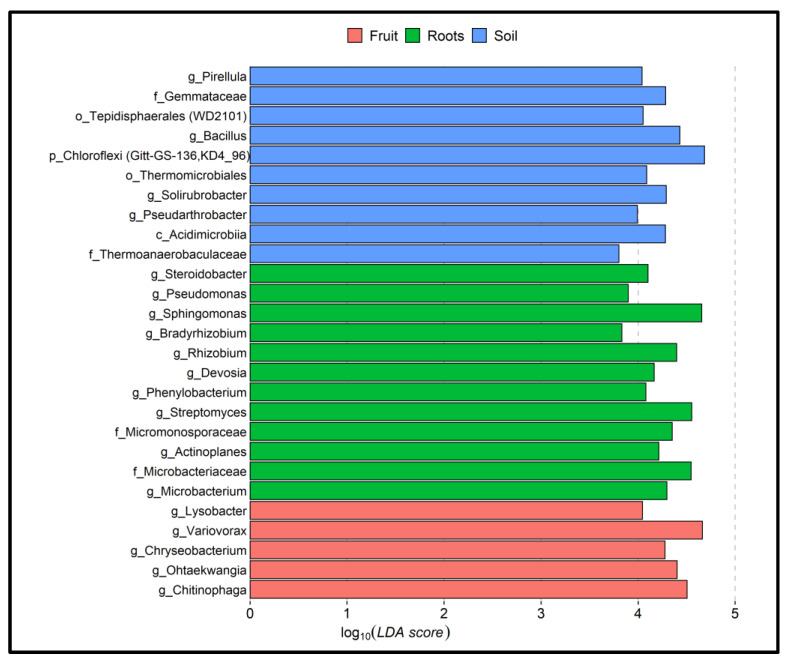
Linear discriminant analysis of taxonomic biomarkers identified in fruit, host plant roots, and rhizosphere samples. The threshold for the LDA score was 3.5.

**Figure 5 jof-08-01062-f005:**
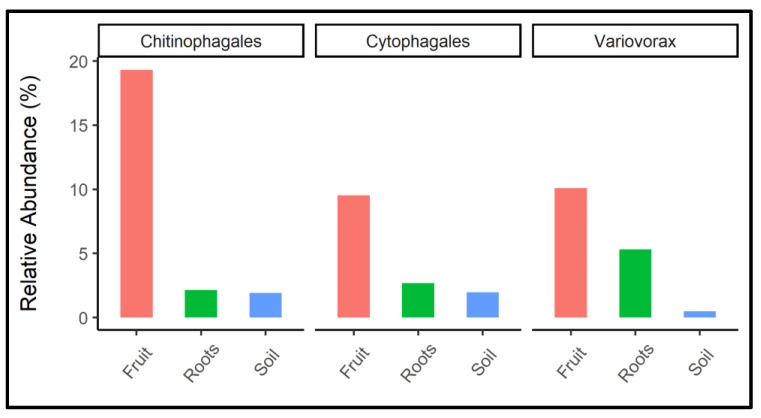
Relative abundance of fruit-associated bacterial groups was found as biomarkers using the LEfSe analysis.

**Table 1 jof-08-01062-t001:** Changes in levels of soil NK in plots of the host plant *Helianthemum sessiliflorum* mycorrhized by *Terfezia boudieri*. The plot was established in 2017 in the dune soil of the Negev Desert. Values are the means of three replicates. NO_2_: Nitrogen dioxide; N.D.: Not detected; NH_4_: Ammonia; and K: Potassium.

Nitrite and Ammonia	2017		2018		2019	
	NO_2_^−^	NH_4_^−^	NO_2_^−^	NH_4_^+^	NO_2_^−^	NH_4_^+^
	(mg/kg)		(mg/kg)		(mg/kg)	
Control-Soil	N.D.	2.3	0.73	-	0.66	4.2
Mycorrhizal-Soil	N.D.	2.4	0.63	-	0.73	12.7
Potassium	K^+^ (mg/L)		K^+^ (mg/L)		K^+^ (mg/L)	
Control-Soil	0.58		8.4		8.3	
Mycorrhizal-Soil	0.41		9.7		20.8	

## Data Availability

Not applicable.

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
