# Peer review of "The Microbiome Structure of the Symbiosis between the Desert Truffle Terfezia boudieri and Its Host Plant Helianthemum sessiliflorum"

_jof, 2022, doi:10.3390/jof8101062_

Round 1
Reviewer 1 Report
The aim of this study was to identify bacteria contained in the fruiting bodies of desert truffle Terfezia boudieri, as well as in the roots of mycorrhizal plants of the Cistaceae family (Helianthemum sessiliflorum) and in the surrounding soil. This study belongs to the direction of the study of the role of prokaryotes in complex symbiotic relationships in different ecosystems, which has been expanding in recent years, and is of interest to researchers in many areas.
The samples were taken over a period of three years in Ramat Negev Desert, Israel. The authors applied modern methods, mainly from the field of molecular genetics and statistical analysis.
The manuscript is well written and illustrated, the results are beyond doubt. Cited literature is cited reasonably.
There are no comments. There is a wish: the authors mentioned that it is planned to study the antimicrobial activity of the isolated microorganisms, among which there are, in particular, actinobacteria, known as the main source of antibiotics. I will wait for the publication of the authors on this topic.
The manuscript may be published in this form.
P.S.
The line 319: Chtinophagales (change to Chitinophagales)
Author Response
PDF attached.

Reviewer 2 Report
This study aimed to identify the rhizosphere microbial community supporting the symbiotic association between the desert truffle and its host plants. There were three compartments, that is fruit, mycorrhized roots and rhizosphere soil. Through sequencing of 16S rRNA gene, this study analyzed the diversity, community composition of the bacterial community between three compartments, then found dominant biomarker bacterial groups in fruit samples by LDA. The results indicated the bacterial of fruit has lower diversity than roots and rhizosphere soil, and has more bacteria which may support plant nutrition and mycelial growth. That may mean the fruit followed a filtering of microbiota. The plants, mycorrhizal fungi and bacteria had a strong tripartite symbiotic association. This association altered soil nutrition and improve growth of truffles in arid condition. So both mycorrhiza and PGPR are necessary inoculation.
1) How long did this experiment last? Is it a long-term positioning test? When was the sampling time? The field management of the experiment plot, sample collection method, comparison of rhizosphere soil and soil collected from the edge of plot may be more detailed.
2) The bacterial community in the fruit is significantly different from that in the rhizosphere soil and roots. But this study doesn’t seem to give some reasons.
3) This study determined soil NPK levels, and thought it related to the tripartite symbiotic association. But this study didn’t describe the relationship of bacterial community and soil nutrient status clearly. In the end, there is a lack of definite inoculation advice. If there is a detailed inoculation recommendation, you can manage another inoculation experiment which contains both mycorrhizal and bacteria. Then the conclusion can be more strong and perfect.
Author Response
PDF attached

Reviewer 3 Report
In this study, bacterial communities associated with the desert truffle Terfezia boudieri and its related plant Helianthemum sessiliflorum were investigated using a metarbacoding approach targeting the 16S rRNA gene fragments. Undoubtely, this work provides significant information to better understand the microbial diversity and their roles in mycorrhizal relationships. However, before publication, there are some issues and edits that should be gave enough attention, as pointed out in the specific comments.
Specific comments:
Line 41: please replace ‘provide’ with ‘provided’
Line 154, line 159, confused, two rarefaction methods were used?
Line 180: ‘soil NPK levels’, I got your meaning, but it is not unoffical.
Line 190-191: according to fig2, three samples in each group, so ‘more than 50% of the samples’ is not good description.
Fig 1, replace ‘Diversity index’ with ‘Shannon index’, please.
Line 212: about the mean values, keep two digits after the decimal point and give SD values.
Line 215: replace ‘nMDS’ with ‘NMDS’, full text unification.
Line 258-259: for the legend of fig3, please modified like this, ‘Relative abundance of dominant bacterial groups at phylum (a) and class (b) levels; Clustering patterns of ….samples (c)’
Line 270-271: please delete the first sentence.
Fig 4, why the LDA analysis was conducted at different taxonomical levels, i.e. genus and family?
Line 285-286: not ‘are’ and ‘include’, ‘were’ and ‘included’ are better.
Line 289-301: be careful to ‘NPK’, ‘NK”, unofficial, moreover, lack ion valence for NO3 etc. Did you conduct a statistical analysis (i.e. significant test) with these values in Table 1?
Line 316-318: can’t found this conclusion from fig2, unless you provide similarity index.
Line 323-327: please refine these three sentences, Cytophagales and Variovorex were already mentioned above, here, you don’t need more words to descibe them as second or third enriched groups.
Line 353-354: the meaning of this sentence is confused, I can’t get any ideas.
Line 358: whether the previous study [65] used cultural methods? please specific.
Line 370: delete Table1, alternatively, you should show your evidences.
Line 374:“This indicates” should be better.
Line 392: wrong words in italic type.
Author Response
PDF attached

Round 2
Reviewer 2 Report
The authors have revised the manuscript according to last comments and the MS has been improved a lot. In my opinion, it can be accepted after some minor revision.
Comments:
1. Line 147. How to classify OTUs? The methods should be described more detail at least including the percentage similarity you used, such as 97% or others.
2. Line 187. NPK should be changed to “N, P and K”. And if it is the first time they occurred in this MS, the abbreviation should be described in full name.
3. How the LDA analysis was conducted? I can’t see the detail in material and methods parts.
4. Where is the results of soil P conditions? You determined it and present the importance of soil P in introduction part. But the data was disappeared.
Author Response
File attached.
